# The Minimum Lindley Lomax Distribution: Properties and Applications

Sadaf Khan [1], Gholamhossein G. Hamedani [2], Hesham Mohamed Reyad [3], Farrukh Jamal [1,*], Shakaiba Shafiq [1] and Soha Othman [4]

1    Department of Statistics, The Islamia University of Bahawalpur, Bahawalpur 63100, Pakistan; smkhan6022@gmail.com (S.K.); shakaiba.hashmi@gmail.com (S.S.)
2    Department of Mathematics, Statistics and Computer Science, Marquette University, Milwaukee, WI 53233, USA; gholamhoss.hamedani@marquette.edu
3    Department of Information Systems and Production Management, Qassim University, Buraydah 52571, Saudi Arabia; hesham_reyad@yahoo.com
4    Department of Applied Statistics, Cairo University, Giza 12613, Egypt; soha_othman@yahoo.com
\*    Correspondence: farrukh.jamal@iub.edu.pk

**Abstract:** By fusing the Lindley and Lomax distributions, we present a unique three-parameter continuous model titled the minimum Lindley Lomax distribution. The quantile function, ordinary and incomplete moments, moment generating function, Lorenz and Bonferroni curves, order statistics, Rényi entropy, stress strength model, and stochastic sequencing are all carefully examined as basic statistical aspects of the new distribution. The characterizations of the new model are investigated. The proposed distribution's parameters were evaluated using the maximum likelihood procedures. The stability of the parameter estimations is explored using a Monte Carlo simulation. Two applications are used to objectively assess the new model's extensibility.

**Keywords:** compounding distributions; Lindley distribution; Lomax distribution; stochastic ordering; stress strength model; characterization

## 1. Introduction

Appropriate data modeling is believed to provide greater insight into the data, divulging its properties and allowing for tracking its characteristics. Consequently, there is a potential for developing efficient methods for clearer grasp of real-world occurrences. We developed a coherent model to help meet the aspirations of applied practitioners in a wide range of scientific domains, inspired by the application of theoretical probability models in applied research. Tahir and Nadarajah [1] provided a deep review of novel approaches that can be adopted to develop new generalized classes ("G-classes" for short) of distributions. In parallel to G-classes, Tahir and Cordiero [2] presented a review on compounding univariate distributions, their expansions, and classes to detect anomaly scenarios under series and parallel structures. In the current article, we adopted the approach extensively discussed in Section 7 of [2], by integrating two continuous cumulative distribution functions (cdfs) together. Cordeiro et al. [3] initiated this idea and proposed the Exponential-Weibull distribution. In the same vein, we proposed minimum Lindley Lomax (minLLx) distribution by compounding the Lindley and Lomax distributions.

The Lindley (L) and Lomax (Lx) distributions are indispensable models for characterizing data, notably in engineering, for the replacement and maintenance of various goods, systems, and reliability processes. For the stated reason, researchers have found ample evidence of studies that conformed to these distributions, namely, Ghitany et al. [4], Ramos and Louzada [5], Singh et al. [6], Oguntunde et al. [7], Wei et al. [8], and Elgarhy et al. [9], just to mention a few. It is an intriguing fact that both the Lindley and the Lomax distributions emerged from an extension of the exponential model, which is commonly used

to quantify the lifetime of a process or device. Assume that a system comprises of two sub-systems that are operating in tandem at the same time, and that the system will collapse if the first sub-system falters. Let us assume further that the failure times of subsystems follow the Lindley and Lomax distributions with $Y$ and $Z$ independent variables having cdfs, respectively, as follows

$$G(y) = 1 - \left( \frac{1 + \theta + \theta y}{1 + \theta} \right) e^{-\theta y}, \ y \geq 0, \ \theta > 0$$

$$H(z) = 1 - (1 + \lambda z)^{-\beta}, \ z \geq 0, \ \lambda, \beta > 0.$$

Then, the new arbitrary variable (av) $X = \min(Y, Z)$ will be called the min Lindley Lomax (minLLx) to determine the system's failure mechanism. The cdf of the minLLx av is follows as

$$F(x) = 1 - \frac{e^{-\theta x}}{(1 + \lambda x)^\beta} \left( \frac{1 + \theta + \theta x}{1 + \theta} \right), \qquad x \geq 0, \quad \theta, \lambda, \beta > 0. \tag{1}$$

The probability density function (pdf), survival function (sf), and hazard rate function (hrf) in harmony with Equation (1) are given, respectively, by

$$f(x) = \frac{e^{-\theta x}}{(1 + \theta)(1 + \lambda x)^{\beta+1}} \left[ \lambda \beta (1 + \theta + \theta x) + \theta^2 (1 + x)(1 + \lambda x) \right], \ x > 0, \ \theta, \ \lambda, \ \beta > 0, \tag{2}$$

$$S(x) = \frac{e^{-\theta x}}{(1 + \lambda x)^\beta} \left( \frac{1 + \theta + \theta x}{1 + \theta} \right)$$

and

$$h(x) = \frac{\lambda \beta (1 + \theta + \theta x) + \theta^2 (1 + x)(1 + \lambda x)}{(1 + \lambda x)(1 + \theta + \theta x)}, \qquad x > 0. \tag{3}$$

From now on, an av $X \sim$ minLLx $(\theta, \lambda, \beta)$ with a pdf is defined by Equation (2).

The purpose of this research is to present and explore the mathematical configurations of a newly developed three-parameter distribution, the minimum Lindley Lomax model, in the perspective of compounding. The rest of the article is composed of seven main components. The minLLx model's essential mathematical features are examined in Section 2. Specific characterizations of the new distribution are pursued in Section 3. The minLLx model's maximum likelihood estimates and observed information matrix are established in Section 4. In Section 5, a simulation study is carried out. Two applications are provided in Section 6. Eventually, in Section 7, there are some closing remarks.

## 2. Structural Properties

The standard mathematical characteristics of the newly suggested minLLx distribution, as stipulated by the cdf in Equation (1), are explored in this phase. In each subcategory, we report a few explicit results.

### 2.1. Quantile Function

Let the $p$th quantile of the minLLx distribution, say $x_p$, is demarcated by $F(x_p) = p$, such that $0 < p < 1$. Then the root of

$$x_p = \frac{1}{\lambda} \left\{ \left[ \frac{(1 + \theta)(1 - p)e^{\theta x_p}}{1 + \theta + \theta x_p} \right]^{-1/\beta} - 1 \right\}. \tag{4}$$

### 2.2. The Shape of the minLLx Distribution

Mathematically, the forms of the minLLx distribution's density and hazard functions can be defined. The acute points of the density function are the roots of the following equation:

$$\frac{-\lambda(1+\beta)}{1+\lambda x} + \left\{ \frac{\theta[\lambda\beta + 2\theta(1+\lambda x)]}{\lambda\beta(1+\theta+\theta x) + \theta^2(1+x)(1+\lambda x)} \right\} = 0.$$

Furthermore, the acute points of the hazard function are the roots of the following equation:

$$\left\{ \frac{\theta[\lambda\beta + 2\theta(1+\lambda x)]}{\lambda\beta(1+\theta+\theta x) + \theta^2(1+x)(1+\lambda x)} \right\} - \frac{\lambda}{1+\lambda x} - \frac{\theta}{1+\theta+\theta x} = 0.$$

The density and hazard functions are visualized in Figures 1 and 2, respectively. The density function has a reverse-J and right-skewed shape with different peeks, while hrf can sometimes be a monotonic (increasing or decreasing), non-monotonic (bathtub), or constant in shape. The standard L and Lx statistical distributions can only create two shapes, whereas the minLLx model can produce a wide number of shapes based on the power parameter beta.

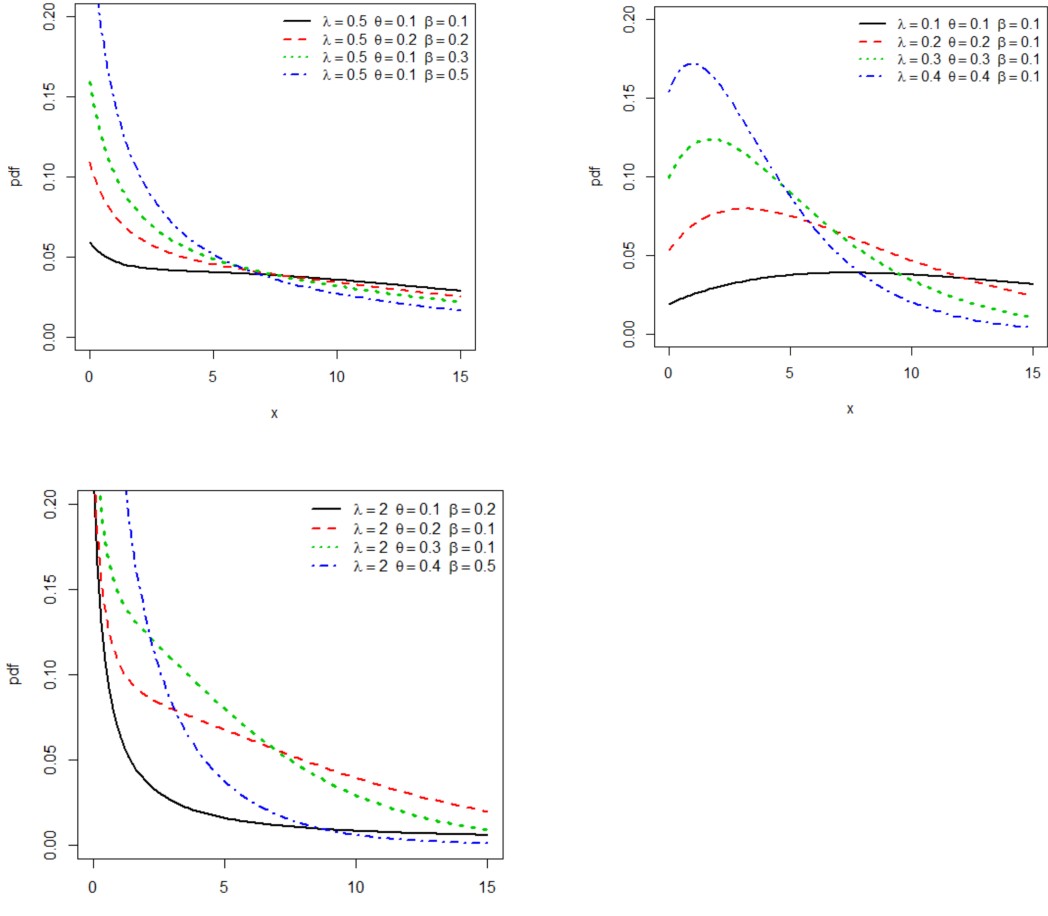

**Figure 1.** Possible figures of the minLLx pdf for parameter values chosen at random.

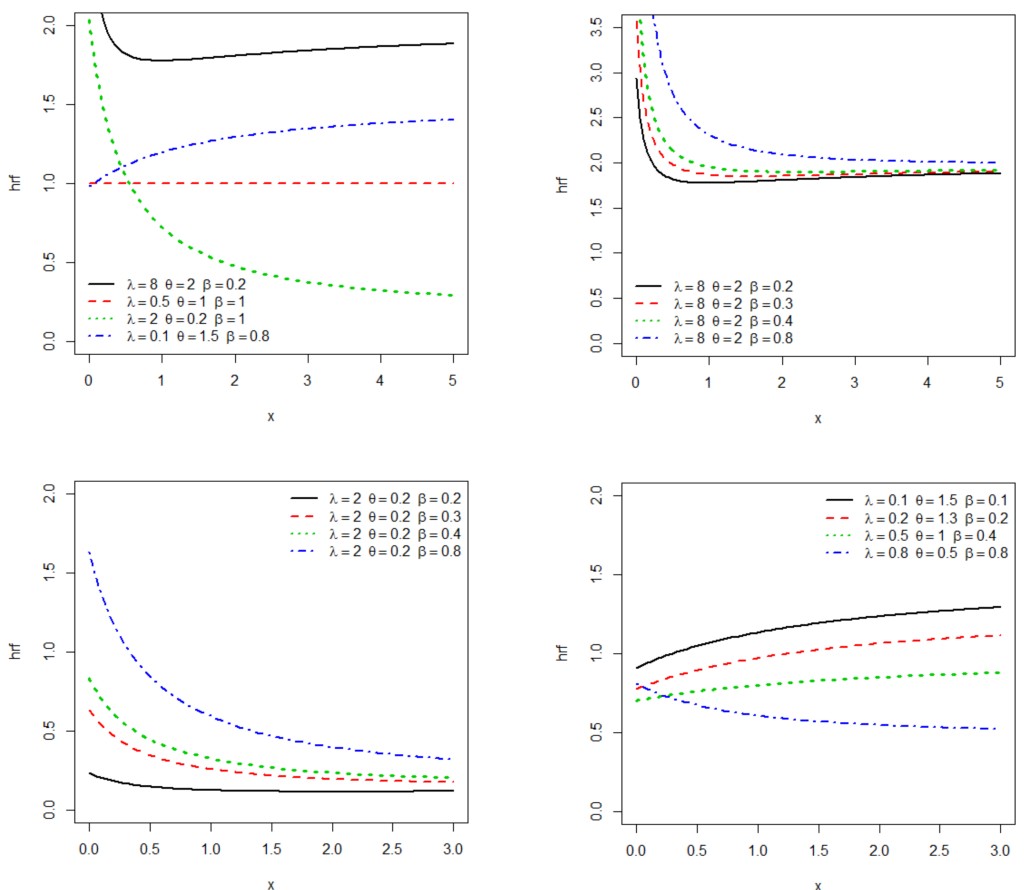

**Figure 2.** Possible figures of the minLLx hrf for parameter values chosen at random.

*2.3. Moments and Moment Generating Function*

Let $X$ be an av with the minLLx distribution, then the ordinary moment, say $\mu'_r$, is given by

$$
\begin{aligned}
\mu'_r = E(X^r) &= \int_{-\infty}^{\infty} x^r f(x)\, dx \\
&= \frac{\lambda\beta}{1+\theta} \int_0^{\infty} x^r(1+\theta+\theta x)(1+\lambda x)^{-\beta-1} e^{-\theta x} dx + \frac{\theta^2}{1+\theta} \int_0^{\infty} x^r(1+x)(1+\lambda x)^{-\beta} e^{-\theta x} dx \\
&= \sum_{j=0}^{\infty} \binom{-\beta-1}{j} \frac{\lambda^{j+1}\beta}{1+\theta} \int_0^{\infty} x^{r+j}(1+\theta+\theta x)e^{-\theta x} dx + \sum_{j=0}^{\infty} \binom{-\beta}{j} \frac{\theta^2\lambda^j}{1+\theta} \int_0^{\infty} x^{r+j}(1+x)e^{-\theta x} dx \\
&= \sum_{j=0}^{\infty} \binom{-\beta-1}{j} \frac{\lambda^{j+1}\beta(r+\theta+j+2)\Gamma(r+j+1)}{(1+\theta)\theta^{r+j+1}} + \sum_{j=0}^{\infty} \binom{-\beta}{j} \frac{\lambda^j(r+\theta+j+1)\Gamma(r+j+1)}{(1+\theta)\theta^{r+j}} \\
&= \sum_{j=0}^{\infty} \frac{\lambda^j \Gamma(r+j+1)}{(1+\theta)\theta^{r+j+1}} \left\{ \lambda\beta(r+\theta+j+2)\binom{-\beta-1}{j} + \theta(r+\theta+j+1)\binom{-\beta}{j} \right\},
\end{aligned}
$$

(5)

where $\Gamma(n) = \int_0^{\infty} x^{n-1} e^{-x} dx$ is the gamma function. Substituting $r = 1, 2, 3, 4$ into (5), we obtain the mean $= \mu'_1$, variance $= \mu'_2 - \mu'^2_1$, *skewness* $= \left\{ \mu'_3 - 3\mu'_2\mu'_1 + 2\mu'^3_1 \right\}^2$ $\left\{ \mu'_2 - (\mu'_1)^2 \right\}^{-3}$ and *kurtosis* $= \left\{ \mu'_4 - 4\mu'_3\mu'_1 + 6\,\mu'_2\mu'^2_1 - 3\mu'^4_1 \right\} \left\{ \mu'_2 - (\mu'_1)^2 \right\}^{-2}$. Table 1 provides the mean, variance, standard deviation, skewness, and kurtosis of $X$ for different combinations of $\theta, \lambda, \beta$ as $A_1 :\ \theta = 3.5,\ \lambda = 0.4,\ \beta = 0.5; A_2 :\ \theta = 0.3, \lambda = 1,\ \beta = 0.8; A_3 :\ \theta = 1.5,\ \lambda = 0.1,\ \beta = 1.5$, and $A_4 : \theta = 0.3, \lambda = 0.5, \beta = 0.3$.

**Table 1.** Moments, variance, standard deviation, skewness and kurtosis of *X* for randomly selected parameter values of minLLx($\theta,\lambda,\beta$).

| $\mu'_r$ | $A_1$ | $A_2$ | $A_3$ | $A_4$ |
|---|---|---|---|---|
| $\mu'_1$ | 1.126862 | 0.3661565 | 0.9344674 | 0.164529 |
| $\mu'_2$ | 1.406321 | 0.2563243 | 1.76341 | 0.1045506 |
| $\mu'_3$ | 1.896405 | 0.2028995 | 5.733562 | 0.07738473 |
| $\mu'_4$ | 2.72086 | 0.172917 | 28.11525 | 0.06041124 |
| Variance | 0.1365036 | 0.1222537 | 0.890181 | 0.07748082 |
| S.D | 0.369464 | 0.349648 | 0.9434941 | 0.2783538 |
| Skewness | 1.137115 | 1.563494 | 2.448468 | 2.289103 |
| Kurtosis | 1.375744 | 1.731832 | 0.84139 | 1.526684 |

The empirical findings from Table 1 allow us to deduce that the skewness is greater than zero, indicating a lack of symmetry of the tails, specifically an elongated right tail. This signifies that the mean and median are pulled to the right. Moreover, kurtosis values are less than three, demonstrating that the distribution is platykurtic.

The *n*th principal moment of the minLLx distribution, say $\mu_n$, can be acquired from

$$
\mu_n = \sum_{r=0}^{n} \binom{n}{r} (-\mu'_1)^{n-r} E(x^r)
$$
$$
= \sum_{r=0}^{n} \sum_{j=0}^{\infty} \binom{n}{r} \frac{(-\mu'_1)^{n-r}\lambda^j \Gamma(r+j+1)}{(1+\theta)\theta^{r+j+1}} \left\{ \lambda\beta(r+\theta+j+2)\binom{-\beta-1}{j} + \theta(r+\theta+j+1)\binom{-\beta}{j} \right\}. \tag{6}
$$

The *r*th incomplete moment of the minLLx distribution, symbolized by $\varphi_s(t)$, is

$$
\varphi_s(t) = \int_{-\infty}^{t} x^s f(x)\, dx
$$
$$
= \sum_{i=0}^{\infty} \frac{\lambda^i}{(1+\theta)\theta^{s+i+1}} \left\{ \begin{array}{l} \lambda\beta[(1+\theta)\gamma(s+i+1,t)+\gamma(s+i+2,t)]\binom{-\beta-1}{i} \\ +\theta[\theta\gamma(s+i+1,t)+\gamma(s+i+2,t)]\binom{-\beta}{ji} \end{array} \right\}, \tag{7}
$$

where $\gamma(a,x) = \int_0^x t^{a-1} e^{-t}\, dt$ is the lower incomplete gamma function.

The moment generating function, signified by $M_x(t)$, of the minLLx distribution can be acquired as

$$
M_x(t) = E(e^{tx}) = \sum_{j=0}^{\infty} \frac{\lambda^j \Gamma(j+1)}{(1+\theta)\theta^{j+2}} \left\{ \begin{array}{l} \lambda\beta[\theta(\theta+j-t+2)-t]\binom{-\beta-1}{j} \\ +\theta^2(\theta+j-t+1)\binom{-\beta}{j} \end{array} \right\}. \tag{8}
$$

*2.4. Probability Weighted Moments*

Ordinary moments of order statistics are generalized by probability weighted moments of a stochastic process, which naturally arise while dealing with ordinary moments. They also play a significant role in several parametric estimate techniques. The formulation for the probability weighted moments of a chance variable with the minLLx distribution is as follows.

The $(r+s)$th probability weighted moments (PWMs) of a chance variable $X$ with the minLLx distribution, about $M_{r,s}$, follows

$$
\begin{aligned}
M_{r,s} &= E\left(X^r F(x)^s\right) = \int_{-\infty}^{\infty} x^r F(x)^s f(x)dx \\
&= \int_{-\infty}^{\infty} x^r \frac{(1+\lambda x)^{-\beta-1}}{1+\theta}\left\{\lambda\beta(1+\theta+\theta x)+\theta^2(1+x)(1+\lambda x)\right\}e^{-\theta x} \\
&\quad \times \left\{1-(1+\lambda x)^{-\beta}\left(\frac{1+\theta+\theta x}{1+\theta}\right)e^{-\theta x}\right\}^s dx \\
&= \sum_{j=0}^{\infty} \frac{(-1)^j}{(1+\theta)^{j+1}}\binom{s}{j}\int_{-\infty}^{\infty} x^r(1+\lambda x)^{-\beta(j+1)-1}(1+\theta+\theta x)^j e^{-\theta(j+1)x} \\
&\quad \times \left\{\lambda\beta(1+\theta+\theta x)+\theta^2(1+x)(1+\lambda x)\right\}dx \\
&= \underbrace{\sum_{j=0}^{\infty} \frac{(-1)^j\lambda\beta}{(1+\theta)^{j+1}}\binom{s}{j}\int_{-\infty}^{\infty} x^r(1+\lambda x)^{-\beta(j+1)-1}(1+\theta+\theta x)^{j+1}e^{-\theta(j+1)x}dx}_{A} \\
&\quad +\underbrace{\sum_{j=0}^{\infty} \frac{(-1)^j\theta^2}{(1+\theta)^{j+1}}\binom{s}{j}\int_{-\infty}^{\infty} x^r(1+\lambda x)^{-\beta(j+1)}(1+x)(1+\theta+\theta x)^j e^{-\theta(j+1)x}dx}_{B},
\end{aligned}
$$

where

$$
A = \sum_{i=0}^{\infty}\sum_{w=0}^{j+1} \frac{\lambda^i\theta^w(1+\theta)^{j-w-1}\Gamma(r+i+w+1)}{(\theta(j+1))^{r+i+w+1}}\binom{-\beta(j+1)-1}{i}\binom{j+1}{w}
$$

and

$$
B = \sum_{i=0}^{\infty}\sum_{w=0}^{j} \frac{\lambda^i\theta^w(1+\theta)^{j-w}[\theta(j+1)+r+i+w+1]\Gamma(r+i+w+1)}{(\theta(j+1))^{r+i+w+2}}\binom{-\beta(j+1)}{i}\binom{j}{w}.
$$

Consequently, we arrive at

$$
\begin{aligned}
M_{r,s} = \sum_{j,i=0}^{\infty} & \frac{(-1)^j\lambda^i}{(1+\theta)^{j+1}\theta^{r+i}(1+j)^{r+i}}\binom{s}{j} \\
\times & \left\{
\begin{aligned}
&\sum_{w=0}^{j+1} \frac{\lambda\beta(1+\theta)^{j-w-1}\Gamma(r+i+w+1)}{(1+j)^{w+1}}\binom{-\beta(j+1)-1}{i}\binom{j+1}{w} \\
&+\sum_{w=0}^{j} \frac{(1+\theta)^{j-w}[\theta(j+1)+r+i+w+1]\Gamma(r+i+w+1)}{(1+j)^{w+2}}\binom{-\beta(j+1)}{i}\binom{j}{w}
\end{aligned}
\right\}.
\end{aligned} \tag{9}
$$

### 2.5. Order Statistics

The inclusion of sorted random variables, often known as order statistics, is crucial in the modeling of various longevity systems with distinct component structures. David and Nagaraja [10] laid the all-important foundation for this paradigm. The order statistics of the minLLx distribution are linked to having conventional distributional modules; hence their importance is an inarguable fact.

Consider the given scenario as $X_{1:n}\leq X_{2:n},\ldots\leq X_{n:n}$ be the $X_{k:n}$th order statistics corresponding to a sample of size $n$ from the minLLx distribution. The pdf of $X_{k:n}$, the $k$th order statistic, is given by

$$
f_{X_{k:n}}(x) = \frac{1}{\beta(k,n-k+1)}\sum_{w=0}^{n-k}(-1)^w\binom{n-k}{w}f(x)F(x)^{k+w-1}, \tag{10}
$$

where $\beta(.,.)$ is the exact beta function. From (5) and (6), we have

$$
\begin{aligned}
f(x)F(x)^{k+w-1} = \sum_{j=0}^{\infty} \frac{(-1)^j(1+\lambda x)^{-\beta(j+1)-1}e^{-\theta(j+1)x}}{(1+\theta)^{j+1}} \\
\times \left\{ \lambda\beta(1+\theta+\theta x) + \theta^2(1+x)(1+\lambda x) \right\} \begin{pmatrix} k+w-1 \\ j \end{pmatrix}.
\end{aligned}
\tag{11}
$$

Inserting Equation (11) into Equation (10), we have

$$
\begin{aligned}
f_{X_{k:n}}(x) = \sum_{w=0}^{n-k} \sum_{j=0}^{\infty} \frac{(-1)^{w+j}(1+\lambda x)^{-\beta(j+1)-1}e^{-\theta(j+1)x}}{\beta(k,n-k+1)(1+\theta)^{j+1}} \\
\times \left\{ \lambda\beta(1+\theta+\theta x) + \theta^2(1+x)(1+\lambda x) \right\} \begin{pmatrix} n-k \\ w \end{pmatrix} \begin{pmatrix} k+w-1 \\ j \end{pmatrix}.
\end{aligned}
\tag{12}
$$

Furthermore, the *r*th moment of *k*th order statistic for the minLLx distribution is given by

$$
\begin{aligned}
E(x_{k:n}^r) = \sum_{w=0}^{n-k} \sum_{j,i=0}^{\infty} \frac{(-1)^{w+j}\lambda^i\Gamma(r+i+1)}{\beta(k,n-k+1)(1+\theta)^{j+1}(\theta(1+j))^{r+i+1}} \begin{pmatrix} n-k \\ w \end{pmatrix} \begin{pmatrix} k+w-1 \\ j \end{pmatrix} \\
\times \left\{ \lambda\beta(r+\theta+i+2) \begin{pmatrix} -\beta(j+1)-1 \\ i \end{pmatrix} + \theta(1+j)(\theta(1+j)+r+i+1) \begin{pmatrix} -\beta(j+1) \\ i \end{pmatrix} \right\}.
\end{aligned}
\tag{13}
$$

### 2.6. Rényi Entropy

Entropy is a mathematical concept that encapsulates the logical understanding of quantifying various mechanisms. The entropy technique is adaptable in different fields, including bioenergetics, queuing theory, thermodynamics, colligative properties of solutions, and statistics. There are several mechanisms to quantify the entropy of the minLLx distribution. Rényi entropy is established here by subjecting a feasible expression that may be appraised using any analytical software. In the perspective of the minLLx distribution, the following result incorporates a series expansion of this entropy system of measurement.

Rényi entropy is defined as

$$
I_R(X) = (1-\mu)^{-1} \log \int_{-\infty}^{\infty} f(x)^\mu \, dx, \ \mu > 0, \ \mu \neq 0.
$$

Using Equation (6) and after some manipulations, we have

$$
I_R(X) = (1-\mu)^{-1} \log \left\{ \sum_{i,\ell,w=0}^{\infty} \sum_{j=i}^{\infty} \frac{\lambda^{\mu+w-j}\beta^{\mu-j}\Gamma(i+\ell+w+1)}{\theta^{i+w-2j+1}(1+\theta)^{j+\ell}\mu^{i+\ell+w+1}} \begin{pmatrix} \mu \\ j \end{pmatrix} \begin{pmatrix} j \\ i \end{pmatrix} \begin{pmatrix} \mu-j \\ \ell \end{pmatrix} \begin{pmatrix} j-\mu(\beta+1) \\ w \end{pmatrix} \right\}.
\tag{14}
$$

### 2.7. Stochastic Dominance

Across many distinct fields of probability and statistics, stochastic ordering and inequalities are being employed more extensively to examine the comparative behavior. Biometrics, robustness, econometrics, and actuarial sciences are all fields that have developed this presumption. According to Shaked and Shanthikumar [11], an av $X_1$ is said to be smaller than another av $X_2$ in the likelihood ratio order ($X_1 \leq_{lr} X_2$) if $f_1(x)/f_2(x)$ decreases in $x$. The following theorem shows that the minLLx distribution is ordered in likelihood ratio ordering if the appropriate assumptions exist.

**Theorem 1:** *Let* $X_1 \sim \text{minLLx}(\theta_1, \lambda_1, \beta_1)$ *and* $X_2 \sim \text{minLLx}(\theta_2, \lambda_2, \beta_2)$. *If* $\theta_1 = \theta_2$, $\lambda_1 = \lambda_2$ *and* $\beta_1 \geq \beta_2$ (*or if* $\theta_1 = \theta_2$, $\beta_1 = \beta_2$ *and* $\lambda_1 \geq \lambda_2$), *then* $X_1 \leq_{lr} X_2$.

**Proof:** We have

$$\frac{f_1(x)}{f_2(x)} = \frac{(1+\theta_2)(1+\lambda_2 x)^{1+\beta_2} e^{-(\theta_1-\theta_2)x}}{(1+\theta_1)(1+\lambda_1 x)^{1+\beta_1}} \left\{ \frac{\lambda_1 \beta_1 (1+\theta_1+\theta_1 x) + \theta_1^2 (1+x)(1+\lambda_1 x)}{\lambda_2 \beta_2 (1+\theta_2+\theta_2 x) + \theta_2^2 (1+x)(1+\lambda_2 x)} \right\}.$$

Then

$$\log \frac{f_1(x)}{f_2(x)} = -(\theta_1 - \theta_2) - (1+\beta_1)\log(1+\lambda_1 x) + (1+\beta_2)\log(1+\lambda_2 x) + \log\left(\frac{1+\theta_2}{1+\theta_1}\right)$$

$$+ \log\left[\lambda_1\beta_1(1+\theta_1+\theta_1 x) + \theta_1^2(1+x)(1+\lambda_1 x)\right]$$

$$- \log\left[\lambda_2\beta_2(1+\theta_2+\theta_2 x) + \theta_2^2(1+x)(1+\lambda_2 x)\right].$$

If $\theta_1 = \theta_2$, $\lambda_1 = \lambda_2$ and $\beta_1 \geq \beta_2$ or if $\theta_1 = \theta_2$, $\beta_1 = \beta_2$ and $\lambda_1 \geq \lambda_2$, then we have

$$\frac{d}{dx}\log\frac{f_1(x)}{f_2(x)} = \frac{-\lambda_1(1+\beta_1)}{1+\lambda_1 x} + \frac{\lambda_2(1+\beta_2)}{1+\lambda_2 x} + \frac{\theta_1\{\lambda_1\beta_1+\theta_1[1+\lambda_1(1+2x)]\}}{\lambda_1\beta_1(1+\theta_1+\theta_1 x)+\theta_1^2(1+x)(1+\lambda_1 x)}$$

$$- \frac{\theta_2\{\lambda_2\beta_2+\theta_2[1+\lambda_2(1+2x)]\}}{\lambda_2\beta_2(1+\theta_2+\theta_2 x)+\theta_2^2(1+x)(1+\lambda_2 x)} < 0.$$

Resultantly, $f_1(x)/f_2(x)$ declines in $x$ and hence $X_1 \leq_{lr} X_2$. $\square$

*2.8. Stress Strength Model*

Acquired resistance metrics are used in lifetime testing to ascertain a system's durability. The stress-strength parameter, for instance, is based on the likelihood that a framework would work proficiently if the stress concentration will be less than its toughness. In the perspective of the minLLx distribution, the following result exemplifies a primitive outline for this parameter.

Let $X_1$ and $X_2$ be two independent chance variables with minLLx$(\theta_1, \lambda_1, \beta_1)$ and minLLx$(\theta_2, \lambda_2, \beta_2)$ distributions. Then, the stress$-$strength model is given by

$$R = \Pr(X_2 < X_1) = \int_0^\infty f_1(\theta_1, \lambda_1, \beta_1) F_2(\theta_2, \lambda_2, \beta_2)\, dx$$

$$= 1 - \frac{\lambda_1\beta_1}{(1+\theta_1)1+\theta_2)} \underbrace{\int_0^\infty (1+\lambda_1 x)^{-\beta_1-1}(1+\lambda_2 x)^{-\beta_2}(1+\theta_1+\theta_2)(1+\theta_1+\theta_2 x) e^{-(\theta_1+\theta_2)x}\, dx}_{H}$$

$$- \frac{\theta_1^2}{(1+\theta_1)(1+\theta_2)} \underbrace{\int_0^\infty (1+\lambda_1 x)^{-\beta_1}(1+\lambda_2 x)^{-\beta_2}(1+x)(1+\theta_2+\theta_2 x) e^{-(\theta_1+\theta_2)x}\, dx}_{E},$$

where

$$H = \sum_{j,i=0}^\infty \frac{\lambda_1^j \lambda_2^i \Gamma(j+i+1)}{(\theta_1+\theta_2)^{j+i+3}} \left\{ \begin{array}{l} (1+\theta_1)(1+\theta_2)(\theta_1+\theta_2)^2 + (\theta_1+\theta_2)(j+i+1) \\ \times[\theta_2(1+\theta_1)+\theta_1(1+\theta_2)] + \theta_1\theta_2(j+i+1)(j+i+2) \end{array} \right\} \binom{-\beta_1-1}{j}\binom{-\beta_2}{i},$$

and

$$E = \sum_{j,i=0}^\infty \frac{\lambda_1^j \lambda_2^i \Gamma(j+i+1)}{(\theta_1+\theta_2)^{j+i+3}} \left\{ \begin{array}{l} (1+\theta_2)(\theta_1+\theta_2)^2 + (\theta_1+\theta_2)(1+2\theta_2)(j+i+1) \\ +\theta_2(j+i+1)(j+i+2) \end{array} \right\} \binom{-\beta_1}{j}\binom{-\beta_2}{i}.$$

Therefore, the stress$-$strength model for the minLLx distribution is

$$R = 1 - \sum_{j,i=0}^{\infty} \frac{\lambda_1^j \lambda_2^i \, \Gamma(j+i+1)}{(1+\theta_1)(1+\theta_2)(\theta_1+\theta_2)^{j+i+3}} \begin{pmatrix} -\beta_2 \\ i \end{pmatrix}$$

$$\times \left( \begin{array}{c} \lambda_1 \beta_1 \left\{ \begin{array}{c} (1+\theta_1)(1+\theta_2)(\theta_1+\theta_2)^2 + (\theta_1+\theta_2)(j+i+1) \\ \times [\theta_2(1+\theta_1) + \theta_1(1+\theta_2)] + \theta_1\theta_2(j+i+1)(j+i+2) \end{array} \right\} \begin{pmatrix} -\beta_1 - 1 \\ j \end{pmatrix} \\ + \theta_1^2 \left\{ \begin{array}{c} (1+\theta_2)(\theta_1+\theta_2)^2 + (\theta_1+\theta_2)(1+2\theta_2)(j+i+1) \\ + \theta_2(j+i+1)(j+i+2) \end{array} \right\} \begin{pmatrix} -\beta_1 \\ j \end{pmatrix} \end{array} \right). \tag{15}$$

### 3. Characterization Results

This section outlines how to characterize the minLLx distribution in two ways: (i) on the basis of ratio of two truncated moments and (ii) by using the conditional expectation of certain functions of the av. It is worth emphasizing that for the characterization, (i) the cdf need not have a closed form, but instead relies on the solution of a first order differential equation, which serves as a link between the probability and differential equation. We would also like to highlight that due to the nature of minLLx density function, our characterizations may be the only versions available. Further bear in mind that the characterization (i) is stable in the sense of weak convergence (Glanzel [12]). We present our characterizations (i)–(ii) in the following two subsections.

*3.1. Characterizations on the Basis of Two Truncated Moments*

This subsection deals with the characterizations of minLLx distribution based on the ratio of two truncated moments. Our initial characterization employs a theorem of Glanzel [13], see Theorem A1 of Appendix A. The result is robust even if interval $H$ is not closed, whereas the Theorem's constraint is on the interior of interval $H$.

**Proposition 1.** *Let* $X : Omega \to (0, \infty)$ *be a continuous av and let* $q_1 = \left[\lambda\beta(1 + \theta + \theta x) + \theta^2(1+x)(1+\lambda x)\right]^{-1} e^{\theta x}$ *and* $q_2(x) = q_1(x)(1+\lambda x)^{-1}$ *for* $x > 0$. *The av* $X$ *has pdf (2) iff the function* $\psi$ *defined in Theorem 1 is of the expression*

$$\psi(x) = \frac{\beta(1+\beta)^{-1}}{(1+\lambda x)}, \qquad x > 0.$$

**Proof.** Let us presume that the av $X$ has pdf(2), then

$$(1 - F(x)) \, E[q_1(X)|X \geq x] = \frac{(1+\theta)^{-1}}{\lambda \, \beta \, (1+\lambda x)^{\beta}}, \qquad x > 0,$$

and

$$(1 - F(x)) \, E[q_2(X)|X \geq x] = \frac{(1+\theta)^{-1}}{\lambda \, (\beta + 1) \, (1+\lambda x)^{(\beta+1)}}, \qquad x > 0.$$

Furthermore,

$$\psi(x) \, q_1(x) - q_2(x) = -\frac{q_1(x)}{(\beta+1)(1+\lambda x)} < 0, \qquad \text{for } x > 0.$$

Conversely, if $\xi$ is of the above form, then

$$s'(x) = \frac{\psi'(x) \, q_1(x)}{\psi(x) \, q_1(x) - q_2(x)} = \frac{\lambda \, \beta}{(1+\lambda x)}, \quad x > 0,$$

and consequently

$$s(x) = -\log\left\{(1+\lambda x)^{-\beta}\right\}, \qquad x > 0.$$

Now, according to Theorem 1, $X$ has density (2). $\square$

**Corollary 1.** *Let* $X : \Omega \to (0, \infty)$ *be a continuous av and let* $q_1(x)$ *be as in proposition 3.1. The chance variable X has pdf (2) iff there exist functions* $q_2$ *and* $\psi$ *defined in theorem 1 fullfilling the following differential equation*

$$\frac{\psi'(x) \, q_1(x)}{\psi(x) \, q_1(x) - q_2(x)} = \frac{\lambda \beta}{(1 + \lambda x)}, \qquad x > 0.$$

**Corollary 2.** *The general solution of the differential equation in Corollary 1 is*

$$\psi(x) = (1 + \lambda x)^\beta \left[ -\int \lambda \beta (1 + \lambda x)^{-1} \, (1 + \lambda x)^{-1} (q_1(x))^{-(\beta+1)} q_2(x) \, dx + D \right],$$

where $D$ is a constant. It is worth emphasizing that one set of functions satisfying the above differential equation is given in Proposition 1 with $D = 0$. Clearly, there are other triplets $(q_1, q_2, \psi)$ that satisfy constraints of Theorem 1.

*3.2. Characterizations on the Basis of Conditional Expectation of Certain Functions of an Arbitrary Variable*

In this subsection, we employ a single function $\Psi$ of $X$ and characterize the distribution of $X$ in terms of the truncated moment of $\Psi(X)$. The following proposition has already appeared in Hamedani [14], so we will just state it here that it can be used to characterize the minLLx distribution.

**Proposition 2.** *Let* $X : \quad \Omega \to (e, f)$ *be a continuous av with cdf F. Let* $\Psi(x)$ *be a differentiable function on* $(e, f)$ *with* $\lim_{x \to e^+} \Psi(x) = 1$. *Then for* $\delta \neq 1$,

$$E[\Psi(X) | X \geq x] = \delta \, \Psi(x), \qquad x \in (e, f)$$

*iff*

$$\Psi(x) = [1 - F(x)]^{\frac{1}{\delta} - 1}, \qquad x \in (e, f).$$

**Remark 1.** *For* $(e, f) = (0, \infty)$, $\Psi(x) = \frac{e^{-\theta x / \beta}}{(1 + \lambda x)} \left( \frac{1 + \theta + \theta x}{1 + \theta} \right)^{1/\beta}$ *and* $\delta = \frac{\beta}{\beta + 1}$, *Proposition 2. provides a characterization of the minLLX.*

## 4. Maximum Likelihood Estimation

The maximum likelihood estimates (MLEs) and the observed information matrix for the model parameters of the minLLx distribution will be investigated in this section. Let $x_1, x_2, \ldots, x_n$ be a random sample from the minLLx distribution, then the corresponding log-likelihood function is given by

$$\begin{aligned}
\ell = {}& -n \log(1 + \theta) - \theta \sum_{i=1}^{n} x_i - (1 + \beta) \sum_{i=1}^{n} \log(1 + \lambda x_i) \\
& + \sum_{i=1}^{n} \log\{\lambda \beta (1 + \theta + \theta x_i) + \theta^2 (1 + \lambda x_i)(1 + \lambda x_i)\}.
\end{aligned} \tag{16}$$

The modules of the score vector $\nabla \ell = \left( \frac{\partial \ell}{\partial \theta}, \frac{\partial \ell}{\partial \lambda}, \frac{\partial \ell}{\partial \beta} \right)$ are:

$$\frac{\partial \ell}{\partial \theta} = \frac{-n}{1 + \theta} - \sum_{i=1}^{n} x_i + \sum_{i=1}^{n} \left\{ \frac{(1 + x_i)[\lambda \beta + 2\theta(1 + \lambda x_i)]}{\lambda \beta (1 + \theta + \theta x_i) + \theta^2 (1 + \lambda x_i)(1 + \lambda x_i)} \right\}, \tag{17}$$

$$\frac{\partial \ell}{\partial \lambda} = -(1 + \beta) \sum_{i=1}^{n} \left( \frac{x_i}{1 + \lambda x_i} \right) + \sum_{i=1}^{n} \left\{ \frac{\beta(1 + \theta + \theta x_i) + \theta^2 x_i(1 + x_i)}{\lambda \beta (1 + \theta + \theta x_i) + \theta^2 (1 + \lambda x_i)(1 + \lambda x_i)} \right\}, \tag{18}$$

and

$$\frac{\partial \ell}{\partial \beta} = -\sum_{i=1}^{n} \log(1 + \lambda x_i) + \sum_{i=1}^{n} \left\{ \frac{\lambda(1 + \theta + \theta x_i)}{\lambda\beta(1 + \theta + \theta x_i) + \theta^2(1 + \lambda x_i)(1 + \lambda x_i)} \right\}. \quad (19)$$

The MLEs, say $\hat{\Theta} = (\hat{\theta}, \hat{\lambda}, \hat{\beta})$, of $\Theta = (\theta, \lambda, \beta)^T$, can be obtained by equating the system of nonlinear Equations (17)–(19) to zero and solving them concurrently. The components of the observed information matrix $J(\Theta) = \{J_{wv}\}$ (for $w, v = \theta, \lambda, \beta$( of $\Theta = (\theta, \lambda, \beta)^T$ are given in Appendix B.

## 5. Simulation Study

It is very difficult to compare the theoretical performances of the different estimators for the minLLx distribution. Therefore, simulation is needed to compare the performances of the different methods of estimation, mainly with respect to their biases, mean square errors, and variances for different sample sizes. A numerical study is performed using Mathematica (v9) software. A portion of the used codes are provided as Supplementary Materials. Different sample sizes are considered through the experiments at size $n = 50, 100, 200, 300$, and $500$. For the defined sample size $n$, the experimental bias and MSE values are the aggregate of values from $N = 2000$ replicated samples of the different values of parameters $\theta$, $\lambda$ and $\beta$, respectively. Traditionally, qf, which is the inverse of cdf, i.e., $Q(u) = F^{-1}(p) = \min\{x : F(x) \geq p\}$, is employed. However, in this case, it is not possible to obtain the qf of the minLLx distribution unequivocally. To obtain the minLLx variates, instead, we can implement the Newton−Raphosn algorithm as follows:

I.      Set the values for $n$, $\lambda$, $\theta$, and $\beta$, as well as the starting value of $x_0$.
II.      Develop $U \sim Uniform\,(0,1)$.
III.      Update $x_0$ each time via the Newton−Raphson's methodology, as shown below.

$$x_* = x_0 - R(x_0; \lambda, \theta, \beta)$$

where $R(x_0; \lambda, \theta, \beta) = \frac{F(x_0; \lambda, \theta, \beta)}{f(x_0; \lambda, \theta, \beta)}$, and $F(x_0; \lambda, \theta, \beta)$ and $f(x_0; \lambda, \theta, \beta)$ are cdf and pdf (in Equations (1) and (2)) of minLLx distribution, respectively.

I.      If $|x_0 - x_*| \leq \varepsilon$, where $\varepsilon$ is very small tolerance limit, then store $x_0 = x_*$ as a variate from minLLX $(\lambda, \theta, \beta)$ distribution.
II.      If $|x_0 - x_*| \geq \varepsilon$, fix $x_0 = x_*$ and then proceed to step III.
III.      In order to develop $x_1, x_2, x_3, \ldots, x_n$, steps II-V are repeated $n$ times.

The average estimates, biases, MSEs, coverage probabilities (CPs), and confidence intervals (CIs), at 95% and 99%, on the basis of different parameter combinations, are reported in Tables 2–5 respectively.

**Table 2.** The MLEs, Bias, MSE, and CPs for the model parameters of the minLLx distribution based on some initial (Init) values.

| n | Para | Init. | MLE | Bias | MSE | 95% CI | | | 99% CI | | |
|---|---|---|---|---|---|---|---|---|---|---|---|
| | | | | | | CPs | LB | UB | CPs | LB | UB |
| | $\theta$ | 1.5 | 2.554 | 1.054 | 1.250 | 0.99 | 2.451 | 2.657 | 1.00 | 2.448 | 2.793 |
| 50 | $\beta$ | 0.85 | 1.763 | 0.913 | 0.857 | 0.96 | 1.746 | 1.797 | 0.99 | 1.719 | 1.808 |
| | $\lambda$ | 0.72 | 1.334 | 0.614 | 0.889 | 0.92 | 1.309 | 1.395 | 0.97 | 1.288 | 1.443 |
| | $\theta$ | 1.5 | 2.527 | 1.027 | 1.137 | 0.94 | 2.471 | 2.583 | 0.97 | 2.454 | 2.601 |
| 100 | $\beta$ | 0.85 | 1.667 | 0.817 | 0.698 | 0.97 | 1.656 | 1.781 | 0.98 | 1.637 | 1.798 |
| | $\lambda$ | 0.72 | 1.227 | 0.507 | 0.733 | 0.95 | 1.215 | 1.266 | 0.96 | 1.202 | 1.291 |

**Table 2.** *Cont.*

| n | Para | Init. | MLE | Bias | MSE | 95% CI | | | 99% CI | | |
|---|------|-------|-----|------|-----|--------|------|------|--------|------|------|
| | | | | | | CPs | LB | UB | CPs | LB | UB |
| 200 | $\theta$ | 1.5 | 2.495 | 0.995 | 1.024 | 0. 90 | 2.469 | 2.521 | 0.98 | 2.520 | 2.599 |
| | $\beta$ | 0.85 | 1.601 | 0.751 | 0.583 | 0.97 | 1.586 | 1.625 | 0.95 | 1.547 | 1.643 |
| | $\lambda$ | 0.72 | 1.111 | 0.391 | 0.526 | 0.95 | 1.084 | 1.159 | 0.94 | 1.005 | 1.187 |
| 300 | $\theta$ | 1.5 | 1.738 | 0.238 | 0.556 | 0.94 | 1.721 | 1.755 | 1.00 | 1.727 | 1.779 |
| | $\beta$ | 0.85 | 1.229 | 0.379 | 0.273 | 0.96 | 1.189 | 1.242 | 0.97 | 1.147 | 1.267 |
| | $\lambda$ | 0.72 | 0.997 | 0.277 | 0.377 | 0.95 | 0.979 | 1.015 | 0.97 | 0.958 | 1.093 |
| 500 | $\theta$ | 1.5 | 1.712 | 0.212 | 0.484 | 0.96 | 1.701 | 1.723 | 0.98 | 1.694 | 1.754 |
| | $\beta$ | 0.85 | 1.003 | 0.153 | 0.097 | 0.94 | 0.985 | 1.036 | 0.98 | 0.970 | 1.088 |
| | $\lambda$ | 0.72 | 0.837 | 0.117 | 0.114 | 0.96 | 0.826 | 0.877 | 0.99 | 0.811 | 0.893 |

**Table 3.** The MLEs, Bias, MSE, CPs for the model parameters of the minLLx distribution based on some initial (Init) values.

| $n$ | Para | Init. | MLE | Bias | MSE | 95% CI | | | 99% CI | | |
|-----|------|-------|-----|------|-----|--------|------|------|--------|------|------|
| | | | | | | CPs | LB | UB | CPs | LB | UB |
| 50 | $\theta$ | 2.4 | 3.807 | 1.407 | 2.230 | 0.90 | 3.648 | 3.966 | 0.97 | 3.466 | 3.886 |
| | $\beta$ | 0.5 | 1.128 | 0.628 | 0.604 | 0.98 | 0.932 | 1.324 | 0.94 | 0.87 | 1.386 |
| | $\lambda$ | 0.5 | 0.981 | 0.481 | 0.481 | 0.96 | 0.785 | 1.177 | 0.96 | 0.723 | 1.239 |
| 100 | $\theta$ | 2.4 | 3.595 | 1.195 | 1.678 | 0.97 | 3.719 | 3.870 | 0.98 | 3.454 | 3.627 |
| | $\beta$ | 0.5 | 0.967 | 0.467 | 0.398 | 0.94 | 0.575 | 1.359 | 0.99 | 0.451 | 1.483 |
| | $\lambda$ | 0.5 | 0.864 | 0.364 | 0.382 | 0.97 | 0.472 | 1.256 | 0.98 | 0.348 | 1.38 |
| 200 | $\theta$ | 2.4 | 2.753 | 0.353 | 1.888 | 0.94 | 2.721 | 2.786 | 0.99 | 2.503 | 2.597 |
| | $\beta$ | 0.5 | 0.881 | 0.381 | 0.395 | 0.96 | 0.691 | 1.071 | 0.96 | 0.631 | 1.131 |
| | $\lambda$ | 0.5 | 0.722 | 0.222 | 0.199 | 0.97 | 0.532 | 0.912 | 0.97 | 0.472 | 0.972 |
| 300 | $\theta$ | 2.4 | 2.532 | 0.132 | 0.833 | 0.95 | 2.705 | 2.762 | 1.00 | 2.499 | 2.569 |
| | $\beta$ | 0.5 | 0.646 | 0.146 | 0.271 | 0.96 | 0.42452 | 0.867 | 0.98 | 0.354 | 0.938 |
| | $\lambda$ | 0.5 | 0.637 | 0.137 | 0.269 | 0.97 | 0.415 | 0.858 | 0.99 | 0.345 | 0.929 |
| 500 | $\theta$ | 2.4 | 2.518 | 0.118 | 0.270 | 0.96 | 2.506 | 2.531 | 1.00 | 2.537 | 2.577 |
| | $\beta$ | 0.5 | 0.557 | 0.057 | 0.253 | 0.95 | 0.5276 | 0.586 | 0.99 | 0.518 | 0.596 |
| | $\lambda$ | 0.5 | 0.597 | 0.097 | 0.259 | 0.96 | 0.5676 | 0.626 | 1.00 | 0.558 | 0.636 |

From Tables 2 and 3, we deduced that when the postulated model differs significantly from the genuine model, as anticipated, the MSE of the estimators rises. The MSE drops as the sample size is increased and the homogeneity disintegrates. In general, when the kurtosis increases the MSE declines. Likewise, if the asymmetry widens, so does the bias, and vice versa. The bias lessens as the kurtosis increases. Therefore, it is evident that as sample size n gets larger, the MSEs and biases reduce. Similarly, the CPs of the confidence interval seems to be quite near to the conventional levels of certainty (95% and 99%), which endorses the already established empirical findings. In a nutshell, we may infer that MLEs perform impressively in estimating the parameters of the minLLx distribution.

**Table 4.** The MLEs, Bias, MSE, and CPs for the model parameters of the minLLx distribution based on some initial (Init) values.

| n | Para | Init. | MLE | Bias | MSE | 95% CI | | | 99% CI | | |
|---|------|-------|-----|------|-----|--------|------|------|--------|------|------|
| | | | | | | CPs | LB | UB | CPs | LB | UB |
| 50 | $\theta$ | 2.4 | 3.551 | 1.151 | 1.575 | 0.90 | 3.648 | 3.966 | 1.00 | 3.466 | 3.886 |
| | $\beta$ | 0.15 | 0.667 | 0.517 | 0.477 | 0.99 | 0.471 | 0.863 | 0.94 | 0.409 | 0.925 |
| | $\lambda$ | 1.5 | 2.778 | 1.278 | 1.883 | 0.92 | 2.582 | 2.974 | 0.97 | 2.52 | 3.036 |
| 100 | $\theta$ | 2.4 | 3.295 | 0.895 | 1.051 | 0.98 | 3.719 | 3.870 | 0.96 | 3.454 | 3.627 |
| | $\beta$ | 0.15 | 0.546 | 0.396 | 0.337 | 0.97 | 0.154 | 0.938 | 0.98 | 0.03 | 1.062 |
| | $\lambda$ | 1.5 | 2.337 | 0.837 | 0.951 | 0.94 | 1.945 | 2.729 | 0.99 | 1.821 | 2.853 |
| 200 | $\theta$ | 2.4 | 3.016 | 0.616 | 0.629 | 0.96 | 2.721 | 2.786 | 0.95 | 2.503 | 2.597 |
| | $\beta$ | 0.15 | 0.881 | 0.731 | 0.784 | 0.96 | 0.691 | 1.071 | 0.97 | 0.631 | 1.131 |
| | $\lambda$ | 1.5 | 1.836 | 0.336 | 0.263 | 0.95 | 1.646 | 2.026 | 0.97 | 1.586 | 2.086 |
| 300 | $\theta$ | 2.4 | 2.842 | 0.442 | 0.345 | 0.97 | 2.705 | 2.762 | 0.98 | 2.499 | 2.569 |
| | $\beta$ | 0.15 | 0.646 | 0.496 | 0.496 | 0.96 | 0.425 | 0.867 | 0.99 | 0.354 | 0.938 |
| | $\lambda$ | 1.5 | 1.772 | 0.272 | 0.324 | 0.95 | 1.551 | 1.993 | 0.97 | 1.480 | 2.064 |
| 500 | $\theta$ | 2.4 | 2.537 | 0.137 | 0.27 | 0.95 | 2.506 | 2.531 | 0.98 | 2.537 | 2.577 |
| | $\beta$ | 0.15 | 0.557 | 0.407 | 0.416 | 0.96 | 0.5276 | 0.5864 | 0.99 | 0.5183 | 0.5957 |
| | $\lambda$ | 1.5 | 1.606 | 0.106 | 0.261 | 0.95 | 1.5766 | 1.6354 | 0.98 | 1.5673 | 1.6447 |

**Table 5.** The MLEs, Bias, MSE, and CPs for the model parameters of the minLLx distribution based on some initial (Init) values.

| $n$ | Para | Init. | MLE | Bias | MSE | 95% CI | | | 99% CI | | |
|---|------|-------|-----|------|-----|--------|------|------|--------|------|------|
| | | | | | | CPs | LB | UB | CPs | LB | UB |
| 50 | $\theta$ | 2.4 | 3.851 | 1.451 | 2.355 | 0.99 | 3.648 | 3.966 | 1.00 | 3.466 | 3.886 |
| | $\beta$ | 0.15 | 0.767 | 0.617 | 0.631 | 0.93 | 0.571 | 0.963 | 0.94 | 0.509 | 1.025 |
| | $\lambda$ | 3.5 | 4.708 | 1.208 | 1.709 | 0.98 | 4.512 | 4.904 | 0.92 | 4.45 | 4.966 |
| 100 | $\theta$ | 2.4 | 3.529 | 1.129 | 1.525 | 0.98 | 3.719 | 3.870 | 0.98 | 3.454 | 3.627 |
| | $\beta$ | 0.15 | 0.665 | 0.515 | 0.515 | 0.97 | 0.273 | 1.057 | 0.95 | 0.149 | 1.181 |
| | $\lambda$ | 3.5 | 4.553 | 1.053 | 1.359 | 0.96 | 4.161 | 4.945 | 0.93 | 4.037 | 5.069 |
| 200 | $\theta$ | 2.4 | 3.119 | 0.719 | 0.767 | 0.98 | 2.721 | 2.786 | 0.94 | 2.503 | 2.597 |
| | $\beta$ | 0.15 | 0.498 | 0.348 | 0.371 | 0.97 | 0.308 | 0.688 | 0.98 | 0.248 | 0.748 |
| | $\lambda$ | 3.5 | 4.078 | 0.578 | 0.584 | 0.96 | 3.888 | 4.268 | 0.99 | 3.828 | 4.328 |
| 300 | $\theta$ | 2.4 | 2.728 | 0.328 | 0.358 | 0.96 | 2.705 | 2.762 | 0.98 | 2.499 | 2.569 |
| | $\beta$ | 0.15 | 0.367 | 0.217 | 0.297 | 0.97 | 0.146 | 0.588 | 0.99 | 0.075 | 0.659 |
| | $\lambda$ | 3.5 | 3.876 | 0.376 | 0.391 | 0.94 | 3.655 | 4.097 | 0.98 | 3.584 | 4.168 |
| 500 | $\theta$ | 2.4 | 2.643 | 0.243 | 0.209 | 0.96 | 2.506 | 2.531 | 0.99 | 2.537 | 2.577 |
| | $\beta$ | 0.15 | 0.268 | 0.118 | 0.164 | 0.95 | 0.2386 | 0.2974 | 0.98 | 0.2293 | 0.3067 |
| | $\lambda$ | 3.5 | 3.711 | 0.211 | 0.195 | 0.95 | 3.6816 | 3.7404 | 1.00 | 3.6723 | 3.7497 |

## 6. Applications

In this portion, we consider two actual cases of the minLLx distribution to showcase its effectiveness. When the pressure is at % anxiety levels, the first data set reflects the failure times of the Kevlar 49/epoxy strands. This data are leptokurtic, unimodal, and substantially right skewed, with a likely outlier (skewness = 3.05 and kurtosis = 14.47). This data set is taken from Andrews and Herzberg [15] and the original source is Barlow et al. [16].The data are: 0.01, 0.01,0.02, 0.02, 0.02, 0.03, 0.03, 0.04, 0.05, 0.06, 0.07, 0.07, 0.08, 0.09, 0.09, 0.10, 0.10, 0.11, 0.11, 0.12, 0.13, 0.18, 0.19, 0.20, 0.23, 0.24, 0.24, 0.29, 0.34, 0.35, 0.36, 0.38, 0.40, 0.42, 0.43, 0.52, 0.54, 0.56, 0.60, 0.60, 0.63, 0.65, 0.67, 0.68, 0.72, 0.72, 0.72, 0.73, 0.79, 0.79, 0.80, 0.80, 0.83, 0.85, 0.90, 0.92, 0.95, 0.99, 1.00, 1.01, 1.02, 1.03, 1.05, 1.10, 1.10, 1.11, 1.15, 1.18, 1.20, 1.29, 1.31, 1.33, 1.34, 1.40, 1.43, 1.45, 1.50, 1.51, 1.52, 1.53, 1.54, 1.54, 1.55, 1.58, 1.60, 1.63, 1.64, 1.80, 1.80, 1.81, 2.02, 2.05, 2.14, 2.17, 2.33, 3.03, 3.03, 3.34, 4.20, 4.69, and 7.89. These data are also used by Cooray and Ananda [17] and Al-Aqtash et al. [18].

The second data set signifies the failure time of 20 components from Murthy et al. [19]. The data are: 0.072, 4.763, 8.663, 12.089, 0.477, 5.284, 9.511, 13.036, 1.592, 7.709, 10.636, 13.949, 2.475, 7.867, 10.729, 16.169, 3.597, 8.661, 11.501, and 19.809.

We obtained the MLEs for the unknown parameters of all competitive models and then compared the results via goodness-of-fit statistics: Anderson-Darling (A*), Cramér-von Mises (W*), AIC (Akaike information criterion), and BIC (Bayesian information criterion). The better model corresponds to the smaller of these criteria. The values for the Kolmogorov Smirnov (KS) statistic and its p-value are also presented.

We compared the minLLx distribution with those of Weibull Lindley (WL) (Asgharzadeh et al. [20]), Lomax (Lx), Lindley (L), quasi Lindley (QL) (Shanker and Mishra [21]), and power Lomax (PLx) (Rady et al. [22]). The MLEs, their standard errors (SEs), and some goodness of fit statistics of the models for the respective data sets are introduced in Tables 6–9. The estimated pdf and cdf plots of all competitive distributions for the two data sets are displayed in Figures 3 and 4, respectively.

**Table 6.** The MLEs alongside their accompanying SEs (in parenthesis) for the first data set.

| Distribution | ML Estimates with SEs | | | | | |
|---|---|---|---|---|---|---|
| | $\hat{\lambda}$ | $\hat{\beta}$ | $\hat{\theta}$ | $\hat{\alpha}$ | $\hat{a}$ | $\hat{b}$ |
| minLLx | 29.1543 (24.5461) | 1.1967 (0.1353) | 0.0565 (0.0444) | - | - | - |
| WL | - | - | - | 54.8909 (46.5022) | 0.1262 (0.0029) | 1.3776 (0.1066) |
| Lx | - | 0.0649 (0.0730) | - | 16.0324 (11.8945) | - | - |
| L | - | — | - | 1.3848 (0.1068) | - | - |
| QL | - | 16.2215 (18.4297) | - | 1.0312 (0.1876) | - | - |
| PLx | - | - | 49.8009 (55.9286) | - | 0.9381 (0.0842) | 48.6282 (64.3737) |

**Table 7.** Some goodness of fit statistics for the fitted models to the first data set.

| Distribution | Goodness-of-Fit Statistics | | | | | | |
|---|---|---|---|---|---|---|---|
| | −LL | A* | W* | KS | *p*-Value | AIC | BIC |
| minLLx | 101.7467 | 0.73166 | 0.1174 | 0.0751 | 0.6188 | 209.4934 | 217.3388 |
| WL | 103.7773 | 0.8412 | 0.1372 | 0.1069 | 0.1985 | 213.5547 | 221.4001 |

**Table 7.** *Cont.*

| Distribution | Goodness-of-Fit Statistics | | | | | | |
|---|---|---|---|---|---|---|---|
| | −LL | A* | W* | KS | *p*-Value | AIC | BIC |
| Lx | 103.2335 | 1.1543 | 0.2082 | 0.0836 | 0.4803 | 210.4669 | 215.6972 |
| L | 104.6558 | 0.8349 | 0.1377 | 0.1062 | 0.2046 | 211.3115 | 213.9267 |
| QL | 103.5036 | 1.0226 | 0.1796 | 0.0892 | 0.3968 | 211.0071 | 216.2374 |
| PLx | 102.9973 | 1.1376 | 0.2044 | 0.0912 | 0.3694 | 211.9947 | 219.8400 |

**Table 8.** The MLEs alongside their accompanying SEs (in parenthesis) for the second data set.

| Distribution | ML Estimates with SEs | | | | | |
|---|---|---|---|---|---|---|
| | $\hat{\lambda}$ | $\hat{\beta}$ | $\hat{\theta}$ | $\hat{\alpha}$ | $\hat{a}$ | $\hat{b}$ |
| minLLx | 23.2537 (6.2332) | 0.2000 (0.0357) | 0.0176 (0.0242) | - | - | - |
| WL | - | - | - | 0.5063 (0.2646) | 0.0022 (0.0049) | 0.1936 (0.0376) |
| Lx | - | 0.0063 (0.0050) | - | 19.2257 (15.1770) | - | - |
| L | - | - | - | 0.2161 (0.0344) | - | - |
| QL | - | 12.7561 (8.1217) | - | 0.1276 (0.0188) | - | - |
| PLx | - | - | 5.1542 (4.2880) | — | 1.2999 (0.2549) | 77.2599 (64.2934) |

**Table 9.** Some goodness of fit statistics for the models fitted to the second data set.

| Distribution | Goodness-of-Fit Statistics | | | | | | |
|---|---|---|---|---|---|---|---|
| | −LL | A* | W* | KS | *p*-Value | AIC | BIC |
| minLLx | 60.4860 | 0.4993 | 0.0891 | 0.2013 | 0.3319 | 126.1758 | 129.0630 |
| WL | 60.8537 | 0.5622 | 0.0992 | 0.2051 | 0.3237 | 127.7075 | 128.2906 |
| Lx | 62.9558 | 0.9314 | 0.1602 | 0.2484 | 0.1422 | 129.9117 | 131.9032 |
| L | 61.3791 | 0.6909 | 0.1203 | 0.2022 | 0.3298 | 126.9583 | 129.7541 |
| QL | 62.6023 | 0.8804 | 0.1514 | 0.2493 | 0.1396 | 129.2046 | 131.1960 |
| PLx | 62.5202 | 0.9067 | 0.1561 | 0.2315 | 2000 | 131.0405 | 134.0277 |

The values in Tables 7 and 9 clearly show that the minLLx distribution has the smallest values for A*, W*, AIC, BIC, and KS, and the largest *p*-values among all competitive models, compelling it to be chosen as the best model. It is clear from Figures 3 and 4, that the new minLLx distribution provides the best fits for the two data sets.

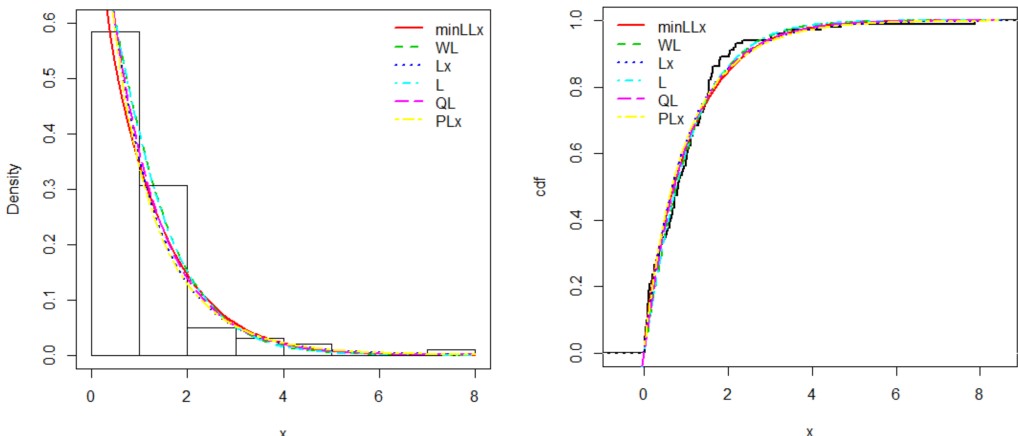

**Figure 3.** Estimated pdf and cdf plots of the minLLx distribution for the first data set.

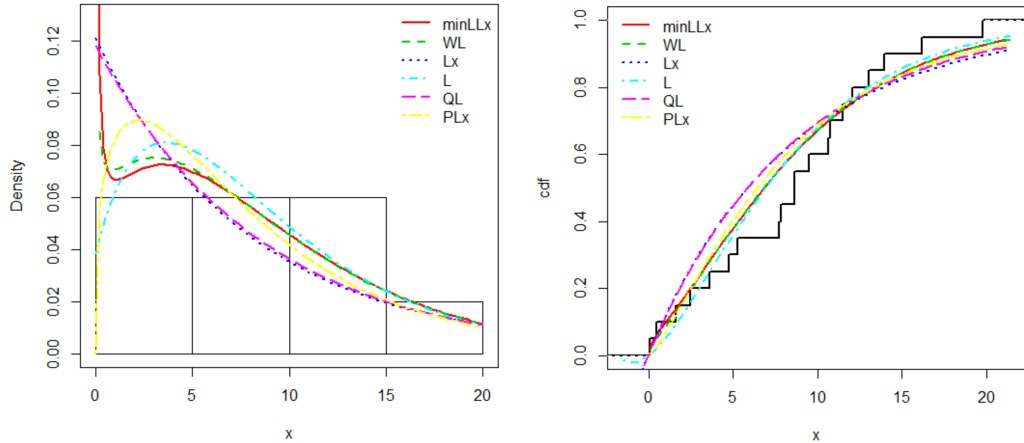

**Figure 4.** Estimated pdf and cdf plots of the minLLx distribution for the second data set.

## 7. Conclusions

By unifying the Lindley and Lomax distributions, we establish a three-parameter distribution called the minimum Lindley Lomax (minLLx). The quantile function, ordinary and incomplete moments, moment generating function, Lorenz and Bonferroni curves, order statistics, Rényi entropy, stress−strength model, and stochastic ordering are all considered as defining attributes of the new model. The envisaged model's characterizations are evaluated. The model parameters are determined using the optimum likelihood criterion, and these projections are assessed using numerical simulations. Two real-world applications exemplify the utility of the new model.

**Supplementary Materials:** Partial codes used in Section 5 are available online at https://www.mdpi.com/article/10.3390/mca27010016/s1.

**Author Contributions:** Conceptualization, S.K. and G.G.H.; methodology, H.M.R.; software, S.O.; validation, F.J., S.K. and H.M.R.; formal analysis, F.J. and H.M.R.; investigation, S.S.; resources, S.O.; data curation, S.S.; writing—original draft preparation, F.J. and G.G.H.; writing—review and editing S.K. and F.J.; visualization, S.K. and S.S.; supervision, S.O.; project administration, F.J. All authors have read and agreed to the published version of the manuscript.

**Funding:** This research received no external funding.

**Conflicts of Interest:** The authors declare no conflict of interest.

## Appendix A

**Theorem A1.** *Let $(\Omega,\ F,\ P)$ be a given probability space and let $H = [a,b]$ be an interval for some $d < b$ ($a = -\infty$, $b = \infty$ might as well be allowed). Let $X : \Omega \to H$ be a continuous av with the distribution function $F$ and let $q_1$ and $q_2$ be two real functions defined on $H$, such that*

$$E[q_2(X)|X \geq x] = E[q_1(X)|X \geq x]\psi(x), \quad x \in H,$$

*is defined with some real function $\eta$. Assume that $q_1$, $q_2 \in C^{-1}(H)$, $\psi \in C^2(H)$ and $F$ is a twice continuously differentiable and strictly monotone function on the set $H$. Finally, assume that the equation $\psi\, q_1 = q_2$ has no real solution in the interior of $H$. Then $F$ is uniquely determined by the functions $q_1$, $q_2$, and $\psi$, particularly*

$$F(x) = \int_a^x C \left| \frac{\psi'(u)}{\psi(u)\, q_1(u) - q_2(u)} \right| \exp(-s(u))\ du \,,$$

*where function $s$ is a solution of the differential equation $s' = \frac{\psi'\, q_1}{\psi\, q_1 - q_2}$ and $C$ is the normalization constant, such that $\int_H dF = 1$.*

We like to mention that this kind of characterization based on the ratio of truncated moments is stable in the sense of weak convergence (see Glanzel [12]), in particular, let us assume that there is a sequence $\{X_n\}$ of avs with a distribution function $\{F_n\}$, such that the functions $q_{1n}$, $q_{2n}$, and $\psi_n$ ($n \in N$) satisfy the conditions of Theorem 1, and let $q_{1n} \to q_1$, $q_{2n} \to q_2$ for some continuously differentiable real functions $q_1$ and $q_2$. Finally, let $X$ be a chance variable with distribution $F$. Under the condition that $q_{1n}(X)$ and $q_{2n}(X)$ are uniformly integrable and the family $\{F_n\}$ is relatively compact, the sequence $X_n$ converges to $X$ in distribution if and only if $\psi_n$ converges to $\psi$, where

$$\psi(x) = \frac{E[q_2(X)|X \geq x]}{E[q_1(X)|X \geq x]}.$$

This stabilization theorem ensures that the precision of the distribution function is duplicated in the subsequent convergence of functions $q_1$, $q_2$, and $\psi_n$. It ensures, e.g., that the characterization on the Wald distribution coincides with that on the Levy-Smirnov distribution if $\alpha \to \infty$. The application of this theorem over certain challenges in analytical techniques, such as the estimation of the parameters of discrete distributions, is yet another corollary of Theorem 1's stability condition. The functions $q_1$, $q_2$, and in particular, $\psi$ should be as straightforward and feasible for this reason. Although the function quartet is not distinctive, it is frequently possible to choose $\psi$ as a linear combination. As a direct consequence, it is worth considering a few specific instances in order to develop innovative characterizations that capture the link between individual continuous univariate distributions and are relevant in other disciplines of science.

## Appendix B

The components of the observed information matrix are the following

$$\frac{\partial^2 \ell}{\partial \theta^2} = \frac{-n}{(1+\theta)^2} + \sum_{i=1}^n \left\{ \frac{\begin{array}{c} 2(1+x_i)(1+\lambda x_i)\left[\lambda\beta(1+\theta+\theta x_i) + \theta^2(1+\lambda x_i)(1+\lambda x_i)\right] \\ -\left[\lambda\beta(1+x_i) + 2\theta(1+x_i)(1+\lambda x_i)\right]^2 \end{array}}{\left[\lambda\beta(1+\theta+\theta x_i) + \theta^2(1+\lambda x_i)(1+\lambda x_i)\right]^2} \right\},$$

$$\frac{\partial^2 \ell}{\partial \theta \, \partial \lambda} = \sum_{i=1}^{n} \left\{ \frac{(1+x_i)\left\{ \begin{array}{c} (\beta + 2\theta x_i)\left[\lambda\beta(1+\theta+\theta x_i) + \theta^2(1+x_i)(1+\lambda x_i)\right] \\ -\left[\lambda\beta + 2\theta(1+\lambda x_i)\right]\left[\beta(1+\theta+\theta x_i) + \theta^2 x_i(1+x_i)\right] \end{array} \right\}}{\left[\lambda\beta(1+\theta+\theta x_i) + \theta^2(1+\lambda x_i)(1+\lambda x_i)\right]^2} \right\},$$

$$\frac{\partial^2 \ell}{\partial \theta \, \partial \beta} = \lambda \sum_{i=1}^{n} \left\{ \frac{\begin{array}{c}(1+x_i)\left[\lambda\beta(1+\theta+\theta x_i) + \theta^2(1+x_i)(1+\lambda x_i)\right] \\ -(1+\theta+\theta x_i)\left[\lambda\beta(1+x_i) + 2\theta(1+x_i)(1+\lambda x_i)\right]\end{array}}{\left[\lambda\beta(1+\theta+\theta x_i) + \theta^2(1+\lambda x_i)(1+\lambda x_i)\right]^2} \right\},$$

$$\frac{\partial^2 \ell}{\partial \lambda^2} = -\sum_{i=1}^{n} \left\{ \frac{\left[\beta(1+\theta+\theta x_i) + \theta^2 x_i(1+x_i)\right]^2}{\left[\lambda\beta(1+\theta+\theta x_i) + \theta^2(1+\lambda x_i)(1+\lambda x_i)\right]^2} \right\},$$

$$\frac{\partial^2 \ell}{\partial \lambda \, \partial \beta} = \sum_{i=1}^{n} \left( \frac{x_i}{1+\lambda x_i} \right) + \sum_{i=1}^{n} \left\{ \frac{\theta^2(1+x_i)(1+\theta+\theta x_i)}{\left[\lambda\beta(1+\theta+\theta x_i) + \theta^2(1+\lambda x_i)(1+\lambda x_i)\right]^2} \right\},$$

$$\frac{\partial^2 \ell}{\partial \beta^2} = -\lambda^2 \sum_{i=1}^{n} \left\{ \frac{(1+\theta+\theta x_i)^2}{\left[\lambda\beta(1+\theta+\theta x_i) + \theta^2(1+\lambda x_i)(1+\lambda x_i)\right]^2} \right\}.$$

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
