# Peer review of "The Minimum Lindley Lomax Distribution: Properties and Applications"

_mca, doi:10.3390/mca27010016_

Round 1

Reviewer 1 Report

Report on the article “The Minmum Lindley Lomax Distribution: Properties and

Applications

MS No: mca-1567566

In this manuscript, the authors studied a new three parameter continuous model called the minmum Lindley Lomax (minLLx) distribution by compounding the Lindley and Lomax distributions. Basic statistical properties of the new distribution are studied including the quantile function, ordinary and incomplete moments, moment generating function, Lorenz and Bonferroni curves, order statistics, Rényi entropy, stress strength model and stochastic ordering. The characterizations of the new model is investigated. The parameters of the proposed distribution are estimated via the method of maximum likelihood. A Monte Carlo simulation is presented to examine the behavior of the parameter estimates. The flexability of the new model is assessed by means of two applications.

In my opinion, the topic is appropriate for publication. The article has some merit for publication in this journal. Methodology and computational work are appreciable. I have some minor comments/suggestions before to recommend for publication of the manuscript. The detailed remarks are pointed out below:

Authors should take more care on typos throughout the manuscripts and make it as easy readability.

Please correct the title as “The Minimum Lindley Lomax Distribution: Properties and Applications”

In Abstract please rewrite sentence “The characterizations of the new model is investigated” as The characterizations of the new model are investigated.

Please correct “flexability” as flexibility.

Please correct through the manuscript “minmum” as minimum.

Introduction of minLLx random variable is not impressive; please explain in detail how to get distribution of minimum of two independent variables. How to get the equation (1) and its statistical background? Is it introduced by the authors or otherwise give the references.

Authors stated that “left-skewed with different peek”. Please check the Figure 1 and state clearly.

Sorry to inform to authors that check the basic formulas like skewness and kurtosis, they are based on central moments. Authors have given in terms of raw moments.  Accordingly check all values of Table 1.

I also doubt the mathematical expressions, please cross check once.

Please rewrite the sentence “The experiment will be repeated 2000 times”.

The authors have given just simulation results in Table 2, these is no discussion on outcome of the results. Please provide detailed discussion on outcome of the results.

 Please cross check “The MLEs and some statistics of the models for the data sets are introduced in Tables (1), (2), (3) and (4) respectively”. Table numbers are not correct. Also check “displayed in Figures 8, 9 and 10 respectively”. Figure numbers are not correct.

Please align the references uniformly.

Author Response

 Reviewer 1
1. Authors should take more care on typos throughout the manuscripts and make it as easy readability.
Reply: Thanks for pointing out; we have now improved the whole manuscript more carefully.
2. Please correct the title as “The Minimum Lindley Lomax Distribution: Properties and Applications”
Reply: We have done the suggested correction in this sentence.
3. In Abstract please rewrite sentence “The characterizations of the new model is investigated” as The
characterizations of the new model are investigated.
Reply: We have done the recommended correction in this sentence.
4. Please correct “flexability” as flexibility.
Reply: Thanks for pointing out; it has been corrected.
5. Please correct through the manuscript “minmum” as minimum.
Reply: Thanks for pointing out; it has been corrected.
6. Introduction of minLLx random variable is not impressive; please explain in detail how to get
distribution of minimum of two independent variables. How to get the equation (1) and its statistical
background? Is it introduced by the authors or otherwise give the references.
Reply: We have rewritten the whole introduction section. The cited issue has been taken care of.
7. Authors stated that “left-skewed with different peek”. Please check the Figure 1 and state clearly.
Reply: Thanks for the keen observation; it was a typing error which has been taken care of.
8. Sorry to inform to authors that check the basic formulas like skewness and kurtosis, they are based on
central moments. Authors have given in terms of raw moments. Accordingly check all the values of
Table 1.
Reply: Thanks for the remark; it was a typing error which has been taken care of. The empirical
findings in Table 1 are hence accordingly verified.
9. I also doubt the mathematical expressions, please cross check once.
Reply: Thanks for the concern; we have rechecked the mathematical expressions again.
10. Please rewrite the sentence “The experiment will be repeated 2000 times”.
Reply: We have rephrased this sentence.
11. The authors have given just simulation results in Table 2, these is no discussion on outcome of the
results. Please provide detailed discussion on outcome of the results.
Reply: We have now added the numerical findings of Table 2 in the form of detail discussion.
12. Please cross check “The MLEs and some statistics of the models for the data sets are introduced in
Tables (1), (2), (3) and (4) respectively”. Table numbers are not correct. Also check “displayed in
Figures 8, 9 and 10 respectively”. Figure numbers are not correct.
Reply: The cited issue has been taken care of, thank you.
13. Please align the references uniformly.
Reply: The mentioned issue has been addressed.

Reviewer 2 Report

  1. some English errors exist. For instance, page 1, line 24,  "flexability" should be " flexibility". page 2, line 45, " it will denoted by"  should be " it will be denoted by" .  
  2.  The simulation is thin.  Please add the performance of the confidence intervals. 
  3. how to generate the random data from this distribution? more details should be given. 

Author Response

 Reviewer 2
1. Some English errors exist. For instance, page 1, line 24, "flexability" should be " flexibility". Page 2,
line 45, “it will denoted by” should be " it will be denoted by" .
Reply: We have done the suggested corrections and improved the overall manuscript.
2. The simulation is thin. Please add the performance of the confidence intervals.
Reply: We have now added the performance of the confidence bounds of parameter estimates at
95% and 99%. The mentioned issue has been addressed completely.
3. How to generate the random data from this distribution? More details should be given.
Reply: Now, we have clearly stated the Newton Raphson methodology being adopted to generate
random samples.

Round 2

Reviewer 1 Report

The authors have responded all the comments/suggestions give by me. Hence manuscript may be accepted in this current form. 

Author Response

Reviewer 1

The authors have responded all the comments/suggestions give by me. Hence manuscript may be accepted in this current form. 

Reply: - We are thankful to the reviewer for his valuable comments, due to this able to improved our manuscript.

Reviewer 2 Report

(1) The authors didn't understand my question #2.  Please add the performance of the coverage of the confidence interval.

(2) Please provide a portion of the codes used on page 13 in the simulation section. 

Author Response

  • The authors didn't understand my question #2.  Please add the performance of the coverage of the confidence interval.

Reply: Thank you for the clarity; we have now added the performance of the coverage of the confidence interval.

  • Please provide a portion of the codes used on page 13 in the simulation section.

Reply: - Thanks for comments, we have added codes as supplementary material.

Round 3

Reviewer 2 Report

The revision is OK.